# Compliance with Preventive Measures and COVID-19 Vaccination Intention among Medical and Other Healthcare Students

**DOI:** 10.3390/ijerph191811656

**Published:** 2022-09-15

**Authors:** Branko Gabrovec, Špela Selak, Nuša Crnkovič, Andrej Šorgo, Katarina Cesar, Mario Fafangel, Mitja Vrdelja, Alenka Trop Skaza

**Affiliations:** 1National Institute of Public Health, Trubarjeva Cesta 2, 1000 Ljubljana, Slovenia; 2Faculty of Natural Science and Mathematics, University of Maribor, Koroška Cesta 160, 2000 Maribor, Slovenia

**Keywords:** immunization, vaccine hesitancy, pandemics, COVID-19 vaccine, mistrust, fear, college students, vaccination

## Abstract

Introduction: The purpose of this study is to evaluate compliance with preventive measures and COVID-19 vaccination acceptance among Slovenian students of healthcare and medicine, identify the predictive socio-demographic factors, establish the possible causes, and propose vaccination strategies and programs in response to the findings. Methods: Data were collected using an online survey as part of a large cross-sectional study of full-time students engaged in higher-level study. The survey took place between 9 February and 8 March 2021. Results: A total of 56.3% of medical and other healthcare students surveyed expressed their intention to receive the vaccine at the earliest opportunity, 22.4% said that they would do so at a later date and 21.3% said that they did not intend to get vaccinated. The medical students surveyed showed a greater readiness to get vaccinated at the earliest opportunity than those studying other healthcare disciplines, men more than women, and single persons more than those in a relationship. Students attending a vocational college, professional higher education or university study program (Bologna first cycle) showed less readiness than other students to get vaccinated as soon as they were able. Conclusion: Our research found that 56.3% of those studying health-related subjects intended to get vaccinated at the earliest opportunity. There were significant differences between medical and other healthcare students, as well as differences resulting from the level of higher education study being undertaken. In addition, the conclusions show that there needs to be a strengthening of communication with students regarding COVID-19 and the importance of vaccination. Lessons that we learned in this pandemic should prepare us for the future. Clearer and more effective communication and education in the future regarding the importance of vaccination is the most effective way of preventing communicable diseases.

## 1. Introduction

The World Health Organization declared the SARS-CoV-2 pandemic on 30 January 2020. In response, governments around the world took drastic measures to limit the spread of the virus, which had a major impact on the life and work of their populations [1].

Slovenia confirmed its first cases of infection on 4 March 2020. At the outset, only non-pharmacological measures were available to protect people against the disease [2]. In the general Slovenian population, higher preventive behavior was found in individuals who experienced greater psychological distress, were more anxious, and expressed greater perceived infectability and germ aversion. Greater compliance with preventive behavior was found among women, those sharing a household with people aged over 65, the elderly, and those who knew somebody who had been infected [1]. In December 2020, a temporary authorization to market the first mRNA vaccine was obtained in the European Union [3].

Vaccinations are the most successful public health measure in history, saving millions of lives and preventing multiple diseases, with large societal and economic benefits [4]. We now have vaccines to prevent more than 20 life-threatening diseases (e.g., diphtheria, tetanus, pertussis, influenza, measles) [5].

Vaccine deployment faces an infodemic (information mixed with fear and rumor) with the rise of misinformation that fills knowledge voids under conditions of uncertainty [6]. Negative attitudes toward vaccines and uncertainty or unwillingness to receive vaccinations are major barriers to managing the COVID-19 pandemic in the long-term [7,8,9] because vaccine hesitancy can have effects on both the individual (greater risk of having the disease) and potentially the community (greater virus transmission) [10].

The healthcare sector has become one of the most important leaders in efforts against COVID-19 [11]. The majority of healthcare professionals have front-line roles during this pandemic and are therefore at higher risk of acquiring viral infections [12]. Because they are considered the most reliable source of information on vaccines [12], they are responsible for providing necessary information regarding vaccine safety and dispelling doubts concerning its use [8].

The COVID-19 pandemic has significantly impacted the lives and education of college students. Medical and healthcare students are regarded as an insightful population that is open-minded, educated, and medically informed. They also represent the future health professionals, who are supposed to respond quickly to public health issues [13]. As medical and healthcare students will be the next generation of professionals working in healthcare institutions and are likely to be exposed to COVID-19 patients, achieving high vaccination coverage rates for COVID-19 in this group is mandatory as they can be used as vaccination role models for the public [14].

Vaccine hesitancy is a public health problem and is also evident among healthcare professionals, even in the midst of a viral pandemic [10]. Several studies showed that participants who were hesitant about the COVID-19 vaccine declared two main reasons for hesitancy: not trusting the vaccine and worrying about its side-effects [11,15,16].

Interestingly, a significant difference in the likelihood of taking the COVID-19 vaccine could be observed between students enrolled in different faculties. In some studies, medical students expressed a higher intent to take the vaccine [10,17,18].

The differences are not present only between medical and healthcare students and students of other disciplines, but between different countries, as well. The acceptance group represented 34.9% of Egyptian medical students while only 13% intended to receive the vaccine as soon as possible [14], 41.39% of medical students from Malta were very likely to take the COVID-19 vaccine [10], almost half (56%) of dental students from Florida, Michigan, and Utah were willing to take the COVID-19 vaccine once approved by the FDA [19], and 34.6 % were willing to vaccinate in Saudi Arabia [20]. In a cross-sectional study in 23 countries, 15% of the participants reported vaccine hesitancy, of whom 4% would outright refuse to accept a COVID-19 vaccine. Physicians were the least hesitant [21]. Acceptance levels are much higher among Italian students (non-medical and medical); 86.1% (633) of them reported that they would choose to have a vaccination for COVID-19 [22] Regarding medical students from India, in response to the statement ‘I am willing to take the COVID-19 vaccine when offered’, 4% (43) ‘disagreed’ and 6.6% (70) were ‘not sure’, which means that the COVID-19 vaccine hesitancy was found in one out of every ten respondents [23]. Students from a large public university in the northwest United States expressed their intention to get a COVID-19 vaccine when available (i.e., 91.64%) [24]. The review of the above shows great diversity showing that reasons for differences are most probably embedded in the affective rather than in the cognitive domain.

The purpose of this study is to evaluate compliance with preventive measures and COVID-19 vaccination acceptance among Slovenian students of healthcare and medicine, identify the predictive socio-demographic factors, establish the possible causes, and propose vaccination strategies and programs in response to the findings.

## 2. Materials and Methods

### 2.1. Procedures and Data Collection

A cross-sectional study using a set of previously tested instruments and ad hoc questions created by the authors was chosen as a method to gain insight into various health and socio-demographic aspects of Slovenian post-secondary students affected by COVID-19-induced closures and suspensions of educational activities at tertiary educational institutions. However, for meta-analysis or reviews, the anonymized primary data of the interest group are also available upon request from the corresponding author of the present study.

Data collection was conducted through a self-reported survey as a part of a large cross-sectional study to determine mental health status and factors which may influence post-secondary students in Slovenia. The study took place between 9 February and 8 March 2021 in the whole territory of the Republic of Slovenia.

Participants were recruited online; the research was conducted through a web-based survey platform (https://www.1ka.si/, accessed on 9 March 2021). Simple random sampling was used and invitation letters to participate in the study were sent to all universities, private faculties, and student organizations with a request to forward the invitation to participate to all their students. Participants were informed about various aspects of the study, including their rights to voluntarily participate and withdraw from the study. Ethical approval to conduct the study was obtained from the National Medical Ethics Committee of the Republic of Slovenia (NMEC), Ministry of Health (No. 0120-48/2021/3).

The survey was conducted in one one-month term; therefore, after ending data collection, we received responses from 5999 full-time students, who partially or fully responded to the questionnaires, which was the final count and which did not change after the data collection period. Due to attrition and deletion of all respondents who did not provide full sets of responses, we ended with datasets of 4661 respondents to be subjects of the present study. The population of students enrolled in tertiary education in Slovenia in the year of the study was about 63,000 full-time students; therefore, we succeeded in collecting data from about 8% of them [25].

From 4661 respondents, responses from medical students (N = 529) and health care students (N = 649) were analyzed. The main goal of the study of medicine is the provision of competencies allowing the students’ specialization to work as physicians in public or private institutions. The diversity of healthcare students is greater because they are trained to enter the world of work as technicians in different sectors of public or private healthcare institutions (Table 1).

### 2.2. Instruments

Besides demographic data, two instruments of interest were chosen from the battery of tests used in a study [1].

The first one was an instrument provisionally named Compliance with a proposed hygienic measures scale. It consists of the following seven items: (1) during the COVID-19 pandemic, I consistently adhere to the usage of surgical protective masks; (2) during the COVID-19 pandemic, I regularly wash my hands with soap and water; (3) during the COVID-19 pandemic, I regularly use hand sanitizers; (4) during the COVID-19 pandemic, I consistently adhere to the proper manner of cough and sneeze hygiene; (5) during the COVID-19 pandemic, I avoid social gatherings with people outside my household; (6) during the COVID-19 pandemic, I avoid having contact with larger groups of people; (7) in public places (e.g., a grocery store), I remind people to adhere to the protective measures if they do not follow them.

The response format was a 7-point Likert scale with anchors at 1—completely not true, 2—not true, 3—partly not true, 4—neither true nor not true, 5—partly true, 6—true, 7—completely true. The content instrument proved to be valid and reliable, with a high degree of internal consistency (Cronbach Alpha min 0.80).

The second instrument was an intention to be vaccinated scale. The respondents were asked to select one of the three selected options, as follows: (1) yes at the first opportunity, (2) yes, but sometime later, (3) no, I have no intention.

### 2.3. Statistical Analysis

Data were analyzed using the statistical software IBM SPSS Version 27 (SPSS Inc., Chicago, IL, USA). Compiled data were processed by means of descriptive statistics, correlation analysis, the Kullbach 2Ȋ test, the chi-square test, the Kolmogorov–Smirnov test, and the Mann–Whitney U test. The content of the questionnaires was found to be valid and reliable with a sufficiently high degree of the questionnaires’ internal consistency (Cronbach’s alpha minimum of 0.81) [26]. The significance level was calculated using the statistical significance value of *p* < 0.05.

## 3. Results

As reported by medical and other healthcare students, the highest scores on the Likert scale (1–7) were recorded for consistent compliance with cough hygiene instructions (M ± SD = 6.62 ± 0.66), the consistent use of protective masks (M ± SD = 6.57 ± 0.84), the frequent washing of hands with soap and water (M ± SD = 6.44 ± 0.89), the consistent maintenance of social distancing (M ± SD = 6.14 ± 1.10), the avoidance of contact with large groups of people (M ± SD = 6.07 ± 1.33), and the avoidance of socializing with people who were at a greater risk of developing serious illness (M ± SD = 6.01 ± 1.20).

A total of 56.3% of the medical and healthcare students surveyed expressed their intention to receive the vaccine at the earliest opportunity, 22.4% said that they would do so at a later date and 21.3% said that they did not intend to get vaccinated (Table 2).

We were then interested in examining the differences between the two groups and the various demographic factors at play.

Table 3 shows that 82.9% (423) of the medical students and 34.7% (218) of those studying other healthcare disciplines intended to get vaccinated at the earliest opportunity. The result of the chi-square test (chi-square = 272.718; *p* < 0.001) showed that there were statistically significant differences in relation to the intention to get vaccinated at the earliest opportunity: medical students expressed a greater readiness to do so than those studying other healthcare disciplines.

As far as education level is concerned, the results show that 78.6% (404) of the survey respondents attending a master’s study program, an integrated study program (Bologna second level), or a doctoral study program (Bologna third level), and 37.8% (236) of the respondents attending a vocational college, a professional college-based undergraduate, or a university study program (Bologna first level) intended to get vaccinated at the earliest opportunity.

The result of the chi-square test (chi-square = 198.408; *p* < 0.001) showed that there were statistically significant differences in relation to the intention to get vaccinated at the earliest opportunity: respondents engaged in a master’s study program, an integrated study program (Bologna second level), or doctoral study program (Bologna third level) were prepared to get vaccinated at the earliest opportunity to a greater degree than those engaged in a vocational college, a professional college-based undergraduate, or a university study program (Bologna first level).

In addition to the course of study, we also found statistically significant differences in relation to gender, with 66.3% of men (136) and 54.3% of women (503) intending to get vaccinated at the earliest opportunity. The result of the Kullbach 2Ȋ test (2Ȋ = 12.639; *p* = 0.013) showed that there were statistically significant differences in relation to the intention to get vaccinated at the earliest opportunity, with male students expressing a greater readiness to do so than female students.

Regarding whether a respondent was single or in a relationship, we found that 61.1% (363) of single students, 50.5% (271) of students in a relationship, and 87.5% (7) of students who gave a different response intended to get vaccinated at the earliest opportunity. The result of the Kullbach 2Ȋ test (2Ȋ = 20.620; *p* = 0.000) showed that there were statistically significant differences in relation to the intention to get vaccinated at the earliest opportunity: single medical and other healthcare students expressed a greater readiness to do so than those who were in a relationship.

Of the respondents, 54.6% (513) had no chronic diseases, while 64.6% (113) of those with one chronic disease and 62.5% (15) of those with more than one chronic disease intended to get vaccinated at the earliest opportunity. We found no statistically significant differences in this population with regard to the intention to get vaccinated at the earliest opportunity (chi-square = 8.190; *p* = 0.085).

## 4. Discussion

In line with the national strategy, Slovenia began vaccinating healthcare professionals and staff coming into direct contact with patients, as well as staff and carers in the care sector, on 27 December 2020. At that point, insufficient quantities of the vaccine were available, which meant that it was not yet accessible to all. Vaccination, therefore, proceeded in accordance with the priorities set out in the strategy.

We conducted the cross-sectional study of students between 9 February and 8 March 2021 using an online questionnaire. At that time, vaccination was not yet freely available [27]. The survey was completed by 4661 students, which was around 8% of the student population in Slovenia (figure taken from the enrolment of students in college and higher education, 2018/2019 academic year). We analyzed 1178 surveys completed by medical and other healthcare students. We opted for this group of students because the vaccination views, knowledge and practices of healthcare professionals (as well as those who will comprise the next generation of healthcare professionals) are very important. Vaccination protects one’s health, enables the healthcare system to operate, protects patient health, and provides an important example to the general population. Despite this, a portion of the professional healthcare population has opted not to get vaccinated if it is not compulsory to do so [28].

The students surveyed in our study displayed a high degree of responsibility and mostly complied with the preventive measures. However, there were differences between the acceptance of these measures. They mostly and easily accepted personal measures, such as wearing masks and using sanitizers. Measures where social distancing is proposed were accepted at the lower level, showing that the pandemic is not only a health problem but a social and societal one as well. The bad news is that they do not act proactively. It is true that at this level they do not have full authority of the profession, but they should act in a proactive way, and not only as passive recipients of the measures in the field of their expertise.

They are often hesitant about vaccination, which means that, for whatever reason, they have decided to defer vaccination until a later date. The World Health Organization regards vaccine hesitancy as one of the biggest threats to public health [29]. Vaccine hesitancy is often linked to doubts concerning safety, although this is by no means a new development. In 1853, Great Britain passed a law mandating vaccination against smallpox, which was opposed by the Anti-Vaccination League. The law was struck off the statute book at the end of the 19th century [30].

More than half of the students expressed an intention to get vaccinated against COVID-19 at the earliest opportunity, more than a fifth were undecided, and 21.3% did not intend to get vaccinated. Similar results were obtained in studies of American dental students [19]. There was a statistically significant difference regarding vaccination intention between medical and other healthcare students, and between master’s/doctoral students and those at an earlier stage of their tertiary education. We can compare the results with those obtained from the study of the COVID-19 vaccination among healthcare workers in Los Angeles between September and October 2020, when the vaccine had not yet been registered. COVID-19 vaccine hesitancy was 4.15 times higher among nursing staff than among doctors. Petravić et al. obtained similar results in the large cross-sectional Slovenian study of healthcare professionals’ opinions on the COVID-19 vaccination. The strongest vaccination intention was expressed by doctors, with 84% responding that they would probably or definitely get vaccinated. This was followed by 82% of medical students and 61% of students of other disciplines. The study was conducted among healthcare professionals and secondary and post-secondary healthcare students between 17 and 27 December 2020 [1,31].

The results of the 18th wave of the Slovenian research on the impact of the pandemic on people’s lives (SI-PANDA study), which took place between 9 and 12 November 2021 on a sample of 1026 adults aged between 18 and 74, showed that 23.4% of respondents did not intend to get vaccinated. Concerns about the side-effects of vaccination and the long-term effect on health, along with the belief that the vaccine was unsafe, were the main reasons why these people did not want to get vaccinated [32].

It is interesting to note that there was a statistically significant difference regarding the decision to get vaccinated; male students were more in favor of vaccination than female students, a finding that has also been produced by other researchers [33].

Regarding the differences in relation to vaccination intention among medical and other healthcare students according to the level of complexity of their study, our research confirmed a difference between the acknowledgment of the importance of vaccination among medical students and students of other healthcare disciplines that depended on the level of higher education being undertaken [1,2].

Despite the fact that several studies have confirmed that the COVID-19 Scale (FCV-19S) is robust and works across all genders and age groups, the lack of research in Slovenia makes it impossible to make a more nuanced comparison. The fact that we did not include questions about influenza vaccination, as those who are regularly vaccinated against influenza tend to be more favorably disposed towards vaccination in general, is a further shortcoming of our study.

## 5. Conclusions

Our research found that 56.3% of those studying health-related subjects intended to get vaccinated at the earliest opportunity, 22.4% did not oppose vaccination and would get vaccinated at a later date and 21.3% did not intend to get vaccinated. There were statistically significant differences between medical and other healthcare students, as well as differences resulting from the level of higher education study being undertaken. The conclusions show that there needs to be a strengthening of communication with students regarding COVID-19 and the importance of vaccination. There needs to be clearer and more effective communication and education in the future regarding the importance of vaccination as the most effective way of preventing communicable diseases. This applies to professionals and the general population.

## Figures and Tables

**Table 1 ijerph-19-11656-t001:** Sample.

	N	Female	Male	Missing Answer
Medical students	529	410 (77.5%)	116 (21.9%)	3 (0.6%)
Healthcare students	649	552 (85.1%)	93 (14.3%)	4 (0.6%)
Total	1178	962 (81.7%)	209 (17.7%)	7 (0.6%)

**Table 2 ijerph-19-11656-t002:** COVID-19 vaccination intention.

	N	Valid Percentage
Yes, I will get vaccinated at the earliest opportunity	641	56.3
Yes, I will get vaccinated at a later date	255	22.4
No, I do not intend to get vaccinated	243	21.3
Total	1139	100

**Table 3 ijerph-19-11656-t003:** Vaccination intention among medical and other healthcare students.

Field of Study	Yes, I Will Get Vaccinated at the Earliest Opportunity	Yes, I Will Get Vaccinated at a Later Date	No, I Do Not Intend to Get Vaccinated	Total
Medicine	423 (82.9%)	58 (11.4%)	29 (5.7%)	510 (100%)
Other healthcare disciplines	218 (34.7%)	197 (31.3%)	214 (34%)	629 (100%)

## Data Availability

Supporting results can be found at: https://www.nijz.si/sites/www.nijz.si/files/datoteke/raziskava_o_dozivljanju_epidemije_covid-19_med_studenti_medicine_in_zdravstvenih_ved.pdf (accessed on 12 March 2022) and https://www.nijz.si/sl/ukrepi-na-podrocju-obvladovanja-siritve-covid-19-s-poudarkom-na-ranljivih-skupinah-prebivalstva (accessed on 12 March 2022).

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
