# Peer review of "Compliance with Preventive Measures and COVID-19 Vaccination Intention among Medical and Other Healthcare Students"

_ijerph, 2022, doi:10.3390/ijerph191811656_

Round 1
Reviewer 1 Report
Reviewer comments:
The review is aiming to assess the compliance with preventive measures and CoVid vaccination intention among students.
The surveys analysis revealed the need for the reinforcement of information, knowledge and importance of the vaccination.
This piece of paper may contribute to awareness of strategies against infectious diseases such as CoVid-19.
However, I do have some suggestions before the publication.
Title: “ Compliance not Complience…”.
Abstract
-Avoid the repetitions of words: There needs on pages 24 and 25.
-I would like the authors to consider the conclusions in the view of the preparedness of his country even the world to the future pandemics. The lessons taken from the pandemics should alert the world to pay attention and be ready in the future.
Introduction
On page 58: …two main reasons: not trusting the vaccine and worrying about its side effects.
Materials and Methods
2.1. Procedures and sample
I suggest Procedures and data collection.
Results and Discussion
-The paragraph from lines 122-129 can be rephrased to make it clear to the readers.
-The chi-squared test results reported through the results section could be better listed or added on the tables.
-On line 246, the SI-PANDA is an abbreviation which need to be explained.
Conclusion
-Page 281-282. In line with my suggestion on the abstract, the importance of vaccination should be communicated to all the publics though the study findings were among students.
-Normally in the conclusion you should not give the references.
Overall, the paper is worthy of publication provide these suggestions and minor changes are made.
26/08/2022
Author Response
Dear Reviewer, thank you for your useful comments and suggestions. I have tried to address them all.

Author Response

(The authors gave the same response as above.)

Round 2
Reviewer 2 Report
Dear Authors,
Thank you for accommodating most of my suggestions. I have read through the revised version and have a few rather minor points:
Abstract – Its conclusion is too general and does not reflect your findings. In the main text, your conclusion consists of two parts – the first is a summarizing of your findings and the second is similar to the one from the abstract. So please include these both parts even in the conclusion of the abstract. In other words, a conclusion should always emphasize your findings and the novelty of your study, not only generally known information (in my opinion :)
Introduction – the last paragraphs should clearly define the aim of your study. In the current version, you somehow mix it with your opinion about what your study will bring/open door to.
Materials and Methods - Instruments – Thank you for adding the paragraph. I still think that accompanying your manuscript with your applied questionnaire as a supplementary file would be a better choice but it is up to you. Am I understanding correctly that you did not use any standardized questionnaire but your own? If so, it would be adequate to include information that you've performed pilot testing and such to increase the credibility of the results through your instrument.
A friendly footnote: It would be good to go through the text patiently so that error phrases like "Click or tap here to enter text" do not appear in it.
Best Regards
Author Response
Dear Reviewer,
thank you again. I find your comments and suggestions helpful for improvement of our paper.
Carelessness and some shortcomings could be addressed before submitting our paper...
Yours sincerely
